Combined effects of waggle dance communication and landscape heterogeneity on nectar and pollen uptake in honey bee colonies

Nürnberger Fabian fabian.nuernberger@uni-wuerzburg.de
Steffan-Dewenter Ingolf
Härtel Stephan
Department of Animal Ecology and Tropical Biology, Bayerische Julius-Maximilians-Universität Würzburg , Würzburg , Germany
Brembs Björn
Electronic publication date: 2017 Jun 7
Publication date: 2017
Volume: 5
Electronic Location ID: e3441
Received 2017 Jan 26; Accepted 2017 May 18
Copyright: ©2017 Nürnberger et al.
Copyright year: 2017
Copyright holder: Nürnberger et al.
License: This is an open access article distributed under the terms of the Creative Commons Attribution License, which permits unrestricted use, distribution, reproduction and adaptation in any medium and for any purpose provided that it is properly attributed. For attribution, the original author(s), title, publication source (PeerJ) and either DOI or URL of the article must be cited.
License URL: https://creativecommons.org/licenses/by/4.0/

Keywords: Apis mellifera, Orientation, Recruitment, Landscape ecology, Foraging behaviour, Floral resource distribution

Funding: German Research Foundation SFB 1047, Project C2 Funding was provided by the German Research Foundation (DFG) in the framework of the Collaborative Research Center 1047-Insect Timing: mechanisms, plasticity and fitness consequences, SFB 1047, Project C2 (ISD and SH). The funders had no role in study design, data collection and analysis, decision to publish, or preparation of the manuscript.

==============================
The instructive component of waggle dance communication has been shown to increase resource uptake of Apis mellifera colonies in highly heterogeneous resource environments, but an assessment of its relevance in temperate landscapes with different levels of resource heterogeneity is currently lacking. We hypothesized that the advertisement of resource locations via dance communication would be most relevant in highly heterogeneous landscapes with large spatial variation of floral resources. To test our hypothesis, we placed 24 Apis mellifera colonies with either disrupted or unimpaired instructive component of dance communication in eight Central European agricultural landscapes that differed in heterogeneity and resource availability. We monitored colony weight change and pollen harvest as measure of foraging success. Dance disruption did not significantly alter colony weight change, but decreased pollen harvest compared to the communicating colonies by 40%. There was no general effect of resource availability on nectar or pollen foraging success, but the effect of landscape heterogeneity on nectar uptake was stronger when resource availability was high. In contrast to our hypothesis, the effects of disrupted bee communication on nectar and pollen foraging success were not stronger in landscapes with heterogeneous compared to homogenous resource environments. Our results indicate that in temperate regions intra-colonial communication of resource locations benefits pollen foraging more than nectar foraging, irrespective of landscape heterogeneity. We conclude that the so far largely unexplored role of dance communication in pollen foraging requires further consideration as pollen is a crucial resource for colony development and health.

Introduction

Communication is a key feature in social insect colonies, and allows them to allocate the colony’s work force effectively to necessary tasks at hand (Hölldobler & Wilson, 2009; Seeley, 1995; Wilson, 1971). An example of this is the recruitment for collaborative foraging, where successful scouts guide idle or unsuccessful nest mates to valuable resource locations (Biesmeijer & De Vries, 2001; Dechaume-Moncharmont et al., 2005; Seeley, 1983). The honey bee waggle dance found in the genus Apis is a unique, highly sophisticated and well-studied recruiting behaviour. Honey bees are also capable of spreading information about the resource environment via dance-independent behaviours, e.g., by offering samples of gathered nectar to nest mates via trophallaxis (Farina, Grüter & Díaz, 2005; Grüter, Acosta & Farina, 2006). The waggle dance, however, does not only provide a motivational component that includes information about the presence and identity of rewarding resources. It also includes the well-known instructive component (Menzel et al., 2011; Von Frisch, 1967). A dancing forager communicates the distance and flight angle relative to the sun’s current azimuth, and hence the relatively precise spatial position of a rewarding food source (Von Frisch, 1967).

Studies on the western honey bee (A. mellifera L.) revealed that waggle dances are highly efficient in recruiting foragers to artificial food sources (Sherman & Visscher, 2002; Von Frisch, 1967) and enable honey bee colonies to concentrate their foraging efforts to the most rewarding resources (Schmickl & Crailsheim, 2004; Seeley, 1986; Seeley, 1995; Seeley, Camazine & Sneyd, 1991). Continuous information exchange about variable resource patches could increase resource uptake rates of honey bee colonies significantly (Donaldson-Matasci & Dornhaus, 2012), and provide fitness advantages (Brown, 1988; Dyer, 2010; Seeley & Visscher, 1988). Dance communication might also allow for selective pollen foraging by allocating the colony worker force to preferred pollen sources (Danner et al., 2016), which enables more consistent exploitation of high-quality resources (Donaldson-Matasci & Dornhaus, 2014). Pollen quality may differ significantly between plant species (Haydak, 1970), and pollen quality and diversity are important factors for honey bee health (Alaux et al., 2017; Alaux et al., 2010; Di Pasquale et al., 2016; Di Pasquale et al., 2013).

Nevertheless, the importance of dance communication for an efficient use of nectar or pollen resources in agricultural landscapes is still unclear. Benefits of spatial information conveyed by the instructive component of waggle dancing for colony fitness were found to be highly dependent on resource density, quality and distribution (Donaldson-Matasci & Dornhaus, 2012; Donaldson-Matasci & Dornhaus, 2014; Dornhaus & Chittka, 1999; Dornhaus & Chittka, 2004; Dornhaus et al., 2006; Okada et al., 2012; Sherman & Visscher, 2002). Dornhaus & Chittka (2004) were able to show a significant effect of dance communication on resource uptake in a complex tropical environment. However, there were no detectable benefits of the instructive component of dance communication in human-modified temperate regions, where distribution of resource patches was less complex (Dornhaus & Chittka, 2004). Within a temperate landscape the benefit of dance communication may change with shifting resource conditions over the seasons (Sherman & Visscher, 2002). Landscapes may differ in the number, proportion and spatial arrangement of different habitat types (Tscharntke et al., 2005), which affects resource distributions. Human-modified temperate landscapes are often dominated by intensively used arable land (Benton, Vickery & Wilson, 2003; Robinson & Sutherland , 2002). They are characterized by few large habitat patches (Tscharntke et al., 2005), including mass-flowering crops that provide plenty of easily available resources (Holzschuh et al., 2016; Westphal, Steffan-Dewenter & Tscharntke, 2003). Such areas have low landscape heterogeneity and form simple resource environments, with easy to find resource patches. This can reduce the value of instructive information exchange between foragers (Beekman & Lew, 2008). In addition, landscapes may contain varying amounts of semi-natural habitats for which pollen foragers show a strong preference (Danner et al., 2016; Steffan-Dewenter & Kuhn, 2003; Steffan-Dewenter et al., 2002). Semi-natural habitats increase the complexity of a resource environment by generating a more heterogeneous landscape. Here mean patch sizes are more variable and generally smaller (Beekman & Ratnieks, 2000; Steffan-Dewenter et al., 2002). Increased heterogeneity and decreased patch size raise the value of instructive information exchange among honey bee foragers (Beekman & Lew, 2008). Accordingly, the dance frequency of honey bee foragers increases with higher proportion of semi-natural habitats (Steffan-Dewenter & Kuhn, 2003).

The design of previous studies may have obscured some beneficial effects of waggle dance communication (Schürch & Grüter, 2014). Honey bee colonies in previous studies remained at the same location during the whole experimentation time. This means that foragers were able to gather and exchange information about resources during phases when communication was not disturbed (Sherman & Visscher, 2002; Dornhaus & Chittka, 2004; Donaldson-Matasci & Dornhaus, 2012; Donaldson-Matasci, DeGrandi-Hoffman & Dornhaus, 2013) and were probably able to profit from this information while dance communication was disrupted.

The aim of our study was to investigate the importance of the information about resource locations conveyed by honey bee dance communication for nectar and pollen foraging success of colonies exposed to landscapes with varying resource heterogeneity. We experimentally disrupted the instructive component of dance communication in honey bee colonies and measured nectar and pollen uptake rates. Unlike previous studies, we performed this in a number of spatially separated human-modified temperate landscapes featuring a variety of levels of complexity and resource availability. For the first time in this context, we used landscape heterogeneity, i.e., heterogeneity in the spatial arrangement of resource patches within a landscape, which describes the complexity of the resource environment on the landscape level and independently from the amount of available resources. We expected that the value of dance communication for colony performance would increase with decreasing resource availability and increasing landscape heterogeneity.

Material and Methods

Study region

The study was conducted in Central Europe, in the vicinity of Würzburg, Germany. Within the study region, simple landscapes, dominated by intensive agriculture, and complex landscapes with a mixture of arable land, woodland, hedgerows, meadows and settlements can be delineated. In order to assess the role of waggle dance communication in different resource environments we selected eight circular landscapes (distances among landscapes ranged from 5.0 to 31.2 km) with differing proportions of intensively used arable land and semi-natural habitats (Table 1). Landscapes were analysed within a radius of 2 km (1265.64 ha area), because mean bee foraging distances under comparable circumstances were shown to lie well within this range (Steffan-Dewenter & Kuhn, 2003), and more than 90% of pollen foraging recruitments advertise patches within this distance to the colony (Danner, Härtel & Steffan-Dewenter, 2014). The experiment took place in late summer 2013 (18th July–18th August 2013).

Table 1 Landscape parameters of the eight selected landscapes for a 2,000 m radius buffer around experimental colonies.

Flower cover is given for the two distinct mapping periods. Means, standard errors and ranges.

Landscape parameter	Mean ± se	Range	
% Semi-natural habitat	7.8 ± 2.5	0.4–16.6	
% Arable land	71.5 ± 5.7	51.1–89.6	
Flower cover—Period A (ha)	23.9 ± 2.9	11.1–33.3	
Flower cover—Period B (ha)	12.5 ± 3.2	3.4–26.0	
Mean patch size (ha)	1.5 ± 0.2	0.8–2.3	

Landscape-level floral resource availability and heterogeneity

Resource availability in each of the eight study landscapes was assessed in two steps. Firstly, we distinguished between habitats that provided noteworthy plant resources for honey bees and those that were unlikely to be utilized for foraging. Resource providing habitats were hedgerows, intensively or extensively used grassland, fallows, meadow orchards, maize fields, sunflower fields, legume fields (including alfalfa, white and red clover and legume mixtures) and non-flowering crop fields (predominantly weeds in beet and cereal fields and vineyards). The relative cover of each habitat type was computed using a geographical information system (Arc-GIS) and digital land use data, which was validated by field inspections. Secondly, we estimated total flower cover on the 2,000 m scale. For this purpose, flower cover was assessed in at least three randomly selected 100 m2 plots in each habitat type that provided measurable amounts of resources (Scheper et al., 2015). Total flower cover was extrapolated by summing estimations of mean flower cover per area multiplied by the relative cover of each habitat type across all habitat types in each landscape. In order to keep track of changes in resource distributions over time, the assessment of the flower cover was done twice. The two discrete timespans for which flower cover was assessed in this study were named period A and period B. Period A lasted from 17th July to 2nd August 2013, while vegetation period B lasted from 3rd August to 18th August 2013. Mean patch size of resource-providing habitats, a configurational measure of landscape complexity, was used as proxy for resource heterogeneity in the landscape. Heterogeneous and more complex resource environments are characterized by small mean patch sizes. Flower cover (resource availability) and mean patch size (landscape heterogeneity) were not correlated significantly (r =  − 0.30; t =  − 1.16, df = 14, p = 0.265).

Study organism

Twenty-four colonies of Apis mellifera carnica were established on 11th July 2013 by making nucleus colonies that were equal in size. Each colony was provided with three fully occupied brood combs, two food combs (Zander measure) and a mated queen. All queens were sister-queens from a professional breeder (Schüler, Münster, Germany). Nucleus colonies were inserted into hive boxes with nine frames. The empty space was filled with two empty combs and two wax sheet frames. Sets of three honey bee colonies were placed in the centre of each study landscape on individual levelled tables.

Disruption of waggle dance communication

The hive box design enabled us to disrupt the instructive component of waggle dance communication using a method following the established approach of Sherman & Visscher (2002) and Dornhaus & Chittka (2004). The hive boxes were placed on levelled tables, and rotation of hives by 90°  allowed for combs to be positioned horizontally, preventing bees from orienting their dances in a specific angle to the gravitational cue. All incoming foragers in rotated hives were forced to enter the hive box via the top frame next to the window, to encourage them to dance there (Dornhaus & Chittka, 2004). Combs were held in place by a tight-fitting slot system that prevented tilting while hive boxes were rotated. In a dark hive without additional cues, dances are performed in random directions and no longer provide consistent spatial information about resource locations (Dornhaus & Chittka, 2004; Sherman & Visscher, 2002; Von Frisch, 1967). The successful disruption of waggle dance orientation on horizontal combs in our experimental hive boxes was confirmed by in-hive video recordings (Fig. S1). Dance orientation on horizontal combs can be re-established if dancers are allowed to see the sun, blue sky or any directional light source (Sherman & Visscher, 2002; Von Frisch, 1967). As an additional treatment we attempted to restore dance orientation by providing a directional light source in form of a closable circular window of 2.5 cm in diameter. However, dance observations revealed that dance orientation could not be fully restored (Fig. S1). Therefore, we do not report results of this treatment.

We analysed groups of colonies with (1) disrupted communication: combs were positioned horizontally and dances were disoriented, in order to investigate the impact of disrupted dance communication; and (2) intact communication: combs were positioned vertically in a dark hive, allowing for unimpaired dances.

All sets of three colonies were moved between the eight landscapes every fourth day during night time. This was repeated seven times, so that each set of colonies was placed in each of the eight landscapes for four days by the end of the experiment. For the statistical analysis each four-day period was regarded as a distinct time step. All communication treatments were randomly re-assigned to the three colonies in each landscape at each time step. All 24 colonies were tested in each treatment and landscape. Minimum distance between consecutive colony locations was ten kilometres to prevent foragers from returning to former colony sites (mean = 19.2 km, SD = ±6.9 km). At the same time, this procedure reduced the value of information about the resource environment that was previously acquired by foragers. This prevented carry-over effects from masking the influence of waggle dance communication on resource uptake. The spatial arrangement of resource patches and of landmarks that could guide workers during foraging flights differed considerably between landscapes. Foragers were shown to perform waggle dances advertising resource patches in up to 4.4 km distance on the first day after moving to a new environment (Danner, Härtel & Steffan-Dewenter, 2014). Due to the methodology, it was not possible to record data on colonies blindly.

Colony development

The presence of the marked queen and brood in the colonies was confirmed every eight days and the total brood area was estimated. In one colony a queen had to be replaced by a reserve sister queen, because she died in the course of the experiment. Data obtained from this colony were not excluded from the presented models, as excluding data did not significantly change model outcomes.

Colony weight

Colony weight change is supposed to reflect resource uptake on colony level. Nectar is the main factor influencing colony weight changes on a daily basis (Meikle et al., 2008; Seeley, 1995). A portable platform balance (Kern EOB35K10) was used to weigh the colonies. Each colony was weighed at the beginning of the experiment and on the first and fourth day at each site. Weighing took place during night time, when all foragers were back in their nest and there was no further resource uptake.

Pollen uptake

The complete pollen forage of each colony was sampled throughout the first day that colonies spent in a new landscape. A total of 192 pollen samples was collected. The pollen was gathered using pollen traps with removable perforated plates (5 mm diameter holes) in front of the colony entrance (Keller, Fluri & Imdorf, 2005). Pollen traps were activated during night time after moving the colonies. Deactivation and pollen collection occurred during the consecutive night, following the weighing of colonies.

The pollen samples were stored in a −20 °C freezer. Later on, pollen samples were vacuum-dried, cleaned from insect parts and other artefacts and weighed to the nearest 0.01 g using a lab scale (Kern Type 430-33). Mean weight-loss (± se) by vacuum drying was 17.5% ± 1.3%.

Statistics

We used linear mixed-effects models in R version 3.3.2 (R Core Team, 2016) with the package lme4 (Bates et al., 2014) to test for effects of instructive dance communication, flower cover and mean patch size, as well as respective interactions, on colony weight change and dry-weight of pollen harvest. Effects on colony weight change were only tested during times when pollen traps were not active. Identity of colony, site and time step were included as random factors in each model to address pseudo-replication and design imbalances. P-values, degrees of freedom and F-values were obtained using the R-package lmerTest using the Satterthwaite approximation for degrees of freedom (Kuznetsova, Brockhoff & Christensen, 2015). Minimum adequate models were identified using ANOVA-tests. P-values of factors that were not included in the minimum adequate models but that were relevant for the hypotheses were calculated by adding the respective factor to the minimum adequate model. As integral part of the main hypothesis, the effects of communication treatments on dependent variables were always shown in the figures, regardless of statistical significance. Data on pollen dry-weight were cubic-root transformed to meet the assumption of normal distribution for linear models. Model residuals were visually inspected for spatial autocorrelation and violation of assumptions of normality and homoscedasticity. See Table 2 for an overview of tested factors and interactions.

Table 2 Results of linear mixed effects models relating colony weight change and dry-weight of pollen harvest to explanatory variables.

n = 8 landscapes, n = 24 colonies, n = 127 colony weight measurements, n = 127 pollen samples.

Explanatory variables	nDF	dDF	F	p	
Colony weight change (g)	
Communication	1	111.93	1.03	0.312	
Flower cover	1	125.84	2.64	0.107	
Mean patch size	1	8.93	10.35	0.011	
Communication × flower cover	1	111.37	0.40	0.528	
Communication × mean patch size	1	111.43	0.01	0.924	
Flower cover × mean patch size	1	114.93	7.25	<0.001	
Communication × flower cover × mean patch	1	111.11	0.39	0.532	
Dry-weight of pollen harvest (g)	
Communication	1	111.99	11.02	0.001	
Flower cover	1	107.25	0.52	0.473	
Mean patch size	1	7.82	8.18	0.022	
Communication × flower cover	1	110.75	0.001	0.977	
Communication × mean patch size	1	110.32	1.28	0.260	
Flower cover × mean patch size	1	97.57	1.96	0.164	
Communication × flower cover × mean patch	1	110.72	1.17	0.282	
Notes.

nDF numerator degrees of freedom

dDF denominator degrees of freedom

Results

Effects of dance communication on foraging success

Mean daily colony weight changes did not differ significantly between colonies with disrupted or intact dance communication (Fig. 1; see Table 2 for statistics). Due to the scarcity of floral resources in late summer in the region of Lower Franconia, Germany, all colonies lost weight over the study period and during most time steps (mean weight change = −37.87 g/day, se = ±6.04 g/day, n = 192; Fig. 1).

Figure 1 Effects of dance communication on mean daily weight change (±se) of honey bee colonies.

Disrupted: colonies with horizontal comb position and disoriented dances; and intact: colonies with non-affected dance communication on vertically positioned combs. ns: p > 0.05.

Rotating the combs to a horizontal position and thereby disrupting dance communication significantly reduced the dry-weight of pollen harvest by 40.25% (Fig. 2; Table 2).

Figure 2 Effects of dance communication on mean dry-weight (±se) of pollen harvest collected by honey bee colonies.

Disrupted: colonies with horizontal comb position and disoriented dances; and intact: colonies with nonaffected dance communication on vertically positioned combs. ∗∗∗:p ≤ 0.001.

Effects of flower cover and mean patch size

Mean flower cover of habitat types ranged from 0–85.2 % (mean = 8.4%; se = ± 1.6%). The flower cover in the studied landscapes varied considerably, both among the eight landscapes and between the two distinct mapping periods (Table 1). In every landscape and during each mapping period we recorded highly rewarding patches of nectar-providing crops like sunflower and legume fields, flower-rich areas promoted by agri-environmental schemes, or flower-rich grasslands. Overall there was no significant effect of flower cover on colony weight change (Fig. 3; Table 2) or dry-weight of pollen harvest (Fig. 4; Table 2).

Figure 3 The relationship between flower cover and mean daily weight change (±se) of honey bee colonies.

For statistics see Table 2.

Figure 4 The relationship between flower cover and mean dry-weight (±se) of pollen collected by honey bee colonies.

For statistics see Table 2.

We used mean patch size in a landscape to define landscape heterogeneity (see Table 1 for patch size range), with higher mean patch size in landscapes with lower heterogeneity. Mean patch size was significantly positively correlated with colony weight change (Fig. 5; Table 2) and dry-weight of pollen harvest (Fig. 6; Table 2). Additionally, flower cover affected the impact of mean patch size on colony weight change, with stronger effects of mean patch size when flower cover was high (Fig. 7; Table 2). There was no significant interaction between flower cover or mean patch size of the studied landscapes and the effect of dance communication on foraging success (Figs. 3–6; Table 2).

Figure 5 The relationship between mean patch size and mean daily weight change (±se) of honey bee colonies.

Regression line fitted with linear model. For statistics see Table 2.

Figure 6 The effect of mean patch size within landscapes on mean dry-weight (±se) of pollen collected by honey bee colonies.

Regression line fitted with linear model. For statistics see Table 2.

Figure 7 The effect of mean patch size on mean daily weight change of honey bee colonies depending on flower cover within the landscape.

Grey area: 95% confidence interval. For statistics see Table 2.

Discussion

In this study we analysed the interplay between the instructive component of dance communication and landscape structure, with regard to colony foraging success. Contrary to our hypothesis, we found that honey bee communication about locations of rewarding floral resources did not promote the nectar intake of bee colonies in temperate agricultural landscapes. The amount of pollen collected in colonies within hives that were rotated in order to disrupt dance orientation was reduced by 40%, indicating an important role of instructive communication in pollen foraging. Our data reveal that the amount of brood reared by a colony which is a main driver of pollen foraging activity was not affected by hive rotations (Fig. S2) but we cannot exclude that the horizontal comb position has further unknown effects on brood rearing behaviour or pollen foraging and storage. Landscape heterogeneity affected nectar and pollen foraging success, but in contrast to our expectation, the benefits of instructive dance communication were not modulated by the complexity ofthe resource environment. Resource availability within the tested landscapes had no direct effects on nectar or pollen foraging success, but altered effects of landscape heterogeneity on nectar foraging success.

It is important to keep in mind that we, and others, disrupted only the instructive information in waggle dance recruitment behaviour. Waggle dances also include information about the presence of rewarding nectar or pollen sources, as well as about their identity (Von Frisch, 1967; Von Frisch, 1968). Dancing foragers are also known to activate idle foragers as well as to reactivate experienced but currently unemployed foragers (Grüter & Farina, 2009), so that dancing generally increases forager recruitment (Gilley, 2014; Von Frisch, 1968). Thus dancing can have a positive effect on resource uptake rates that is unrelated to communication of resource location directions.

In our study, colony weight change was not impacted by manipulation of dance communication, although we deliberately placed colonies in experimentally selected environments where effective communication should offer advantages for foraging success. We tested a number of different landscapes that varied significantly in resource availability and heterogeneity. The study was conducted during late summer, when resources in the study region were generally scarce and colonies lost in weight, but some resource-rich patches were still available and information exchange was expected to be valuable (Okada et al., 2012; Sherman & Visscher, 2002). Additionally, repeatedly moving the colonies to a new environment created an exceptionally short-living resource environment. This forced foragers to repeatedly update information about locations of profitable resources instead of making good use of previously acquired information, which might have masked effects in earlier experiments (Schürch & Grüter, 2014). In contrast to our hypothesis, the high temporal turnover and the spatial heterogeneity of resource patches experienced by foragers did not increase the importance of communication. In temperate landscapes the instructive component of waggle dance communication might only prove to be advantageous for nectar foraging in environments under very specific conditions, like strong intra- or interspecific competition (Donaldson-Matasci & Dornhaus, 2012; Seeley & Visscher, 1988) or during specific seasonal resource distributions (Sherman & Visscher, 2002). While the conditions were deliberately chosen in order to identify the specific conditions under which communication of resource location would be beneficial, it is important to note that these conditions are not representative for the whole flowering period. In early spring, for example, resources would also be scarce but possibly much more patchily distributed in form of few flowering trees and scrubs, which may increase the value of directional dance communication. Additionally, if instructive dance communication does only outweigh dancing costs if advertised resources can be used over extended time periods (Schürch & Grüter, 2014), repeatedly moving colonies every four days prevented us from identifying these long-term benefits. This should be addressed in future field experiments. Contrary to our findings in nectar foraging, our data show that the disruption of instructive dance communication had a strong negative effect on pollen foraging. To our knowledge, only two related studies also investigated the effect of instructive dance communication directly on pollen forage instead of colony weight change (Donaldson-Matasci & Dornhaus, 2012; Donaldson-Matasci & Dornhaus, 2014). However, the studies were restricted to Sonoran Desert scrub and grassland habitats. In these non-temperate landscapes dance communication increased pollen uptake rates independently of resource availability, but only if resource distribution was patchy. Additionally, instructive dance communication also proved to be advantageous, depending on resource conditions (Donaldson-Matasci & Dornhaus, 2012). Due to the study design foragers could make use of information on resource locations gathered before communication was disrupted (Schürch & Grüter, 2014) or ignore available dance information in favour of previously acquired information on resource locations (Grüter & Ratnieks, 2011). Therefore, these studies possibly failed to reveal the actual extent of the effect of dance communication. As colonies in our study were moved to unknown landscapes with considerably different spatial features at the same time at which treatments in individual colonies were changed, we prevented that foragers profited from previously acquired information on resource locations. This allowed us to assess the total benefits of directional dance information under the given conditions. Our findings for temperate landscapes under the conditions of sub-optimal resource availability do not support the hypothesis that resource distribution affects the value of directed dance communication in honey bee colonies. It remains to be confirmed if this is also true when foragers can profit from the directional dance information for a longer period of time, as we only investigated effects on short-term benefits. The fact that in the tested temperate landscapes dance communication always improved pollen foraging, but never nectar foraging, is remarkable. We suspect that this is related to the circumstance that honey bee colonies exploit a higher diversity of plant species for pollen than for nectar (Requier et al., 2015). The identity and diversity of pollen sources may have a strong effect on colony health (Alaux et al., 2010; Di Pasquale et al., 2016; Di Pasquale et al., 2013). Dance communication may allow for a selective and diverse but still effective pollen foraging, but may be less important for effective nectar foraging in temperate landscapes. In fact, it was shown that waggle dance communication affects the composition of pollen forage (Donaldson-Matasci & Dornhaus, 2014). A mechanistic explanation for the differences in our findings between nectar and pollen foraging might be, that pollen foragers are more motivated to follow dances and make use of the instructive component of the dances, e.g., of scouts that advertise novel resource patches. It has been shown that previous experience in the field and in-hive olfactory information affect the way foragers deal with available dance information (Farina, Grüter & Arenas, 2012). In addition, pollen foragers were shown to have a preference for pollen collected from plant species found in semi-natural habitats (Danner et al., 2016) which are generally relatively small, scarce, patchily distributed and probably quickly depleted. Therefore, pollen foragers could profit more from the instructive component of dance communication than nectar foragers that commonly forage in presumably more easy to find mass-flowering crop fields or other floral resources with abundant nectar supply (Beekman & Lew, 2008). Additionally, pollen advertisement in plants can be more limited in time than nectar advertisement and pollen within inflorescences can be rapidly depleted (Herrera, 1990; Stone et al., 1999). High ephemerality of pollen sources and possibly increased competition would increase the benefits of effective communication (Dornhaus & Chittka, 2004; Seeley & Visscher, 1988). We cannot rule out the possibility that additional factors affected pollen foraging activity, as disrupting dance communication coincided with hive rotation (Sherman & Visscher, 2002). It could be argued that rotating the hives affected the brood, brood-provisioning behaviour or brood rearing activity which is known to be strongly correlated with pollen foraging activity (Al-Tikrity et al., 1972; Dreller & Tarpy, 2000; Free, 1967; Pankiw Jr, Page & Kim Fondrk, 1998). While a small proportion of larvae may be malformed, brood rearing in general and egg-laying activity of queens are not known to be affected by horizontal comb position (Chauvin, 1960). In our study the amount of reared brood seemed unaffected by comb position (Fig. S2). We cannot exclude that other components of brood rearing activity are affected by hive rotation and further research on this might help to confirm that indeed disruption of the instructive dance communication caused the observed effects on pollen foraging. However, the random exchange of treatments every four days combined with a considerably longer development time of bee brood minimised possible effects of comb rotation on brood rearing. Although our treatment to control for effects of hive rotation by restoring directed dances on horizontal combs did not work, we therefore conclude that the measured effects of comb orientation on pollen foraging success were most probably due to the disrupted instructive dance information. Incoming pollen was only sampled during the first day within a new environment, in order not to disrupt protein supply and hence brood rearing. To which extent our findings can be extrapolated to longer time periods needs further investigation.

Studies on landscape-related foraging patterns of honey bee colonies are still rare (Couvillon Margaret, Schürch & Francis, 2014; Danner, Härtel & Steffan-Dewenter, 2014; Danner et al., 2016; Härtel & Steffan-Dewenter, 2014). In our study, variation in the generally low resource availability within late summer in temperate landscapes had no direct effect on foraging success. Irrespective of overall resource availability, foragers probably concentrated their efforts on few but most valuable resource patches. However, especially in the most resource rich landscapes, landscape heterogeneity had a strong effect on foraging success. Foraging was most successful in landscapes that contained flower-rich, large and easy to find resource patches, like mass-flowering crop fields. With increasing landscape heterogeneity, i.e., decreasing patch sizes, colony foraging success decreased. Foragers presumably spent less time within the smaller, quickly depleted patches (Cresswell & Osborne, 2004) and hence probably more time on travelling between the scattered patches. This may reduce foraging efficiency (Westphal, Steffan-Dewenter & Tscharntke, 2006).

Conclusions

Although there is an increasing number of theoretical studies and field experiments addressing the possible benefits of the instructive component of waggle dance communication (Beekman & Lew, 2008; Donaldson-Matasci, DeGrandi-Hoffman & Dornhaus, 2013; Donaldson-Matasci & Dornhaus, 2012; Donaldson-Matasci & Dornhaus, 2014; Dornhaus & Chittka, 2004; Dornhaus et al., 2006; Okada et al., 2012; Schürch & Grüter, 2014; Sherman & Visscher, 2002), this study demonstrates that we still lack some essential knowledge regarding its actual relevance on colony level. Even in heterogeneous temperate landscapes and under specific conditions that were expected to increase the benefits of advertisement of resource locations, there were no short-term benefits of instructive dance communication for nectar foraging. In an unknown environment individual search abilities of honey bee foragers and newly established knowledge of resource locations may be sufficient to secure colony foraging success. It is possible that communicating nectar resource locations in temperate landscapes will only provide benefits on the long-term (Schürch & Grüter, 2014), which was prevented in our study. Importantly, our data indicate that, within temperate landscapes, waggle dancing plays a far more important role in pollen foraging than in nectar foraging. As pollen is the major protein source in honey bee hives, dance communication can be expected to have significant effects on colony development and health. This underpins the potential evolutionary advantage of dance communication and suggests that future research should focus more on pollen foraging ecology of honey bees.

Supplemental Information

Figure S1 Mean vector lengths (±se) of waggle dances were used as measure for waggle dance directionality in three initial treatments

Waggle dances were recorded from two honey bee colonies within experimental hive boxes in July–August 2016. Recordings were performed during several days and a number of different weather conditions. Only dances that consisted of more than 5 waggle runs were considered and each first and last waggle run was ignored. Colonies were either left unimpaired (vertical; n = 15), i.e. with vertical comb position, or were rotated, so that combs were positioned horizontally. When combs were in horizontal position, we either provided a light source in form of an opened window of 2.5 cm in diameter on top of the hive box (horizontal + light; n = 26) or kept the combs in dark with closed window (horizontal + dark; n = 25). Mean vector lengths of waggle dances were computed by vector addition of the individual waggle run vectors. A maximum mean vector length of 1 would occur, if runs were directed into the exactly same direction with no variation (maximum directionality). Mean vector length decreases with increasing variation in waggle dance run directions, i.e. decreasing directionality, to a minimum of 0 (no directionality). Treatments differed significantly in mean vector length of waggle dances (Kruskal–Wallis test: Chi 2 = 34.72; df = 2; p < 0.001). Dunn’s test revealed that dances on horizontal combs with ( z =  − 5.59; p =  < 0.001) or without light source ( z =  − 4.96; p =  < 0.001) were significantly less directional than dances on vertical combs, which were highly directional. There was no significant difference in the directionality of dances on horizontal combs with either a light source or without light source ( z = 0.69; p = 0.264).

Click here for additional data file.

Figure S2 Time spent in horizontal comb position had no significant effect on brood rearing activity in honey bee colonies within 24 days

The area of brood cells (open and capped) was estimated for all colonies after 24 days using the Liebefelder method (Imdorf et al. 1987) and correlated with the time colonies had spent in horizontal comb position. Linear model: F1,22 = 1.7245, p = 0.2026.

Reference

Imdorf A, Buehlmann G, Gerig L, Kilchenmann V, and Wille H. 1987. A test of the method of estimation of brood areas and number of worker bees in free-flying colonies.Apidologie 18:137-146. 10.1051/apido:19870204

Click here for additional data file.

Supplemental Information 1 Data on landscape parameters and flower availability for the eight selected landscapes and two vegetation periods

Click here for additional data file.

Supplemental Information 2 Data on colony weight changes and dry-weight of pollen samples

Click here for additional data file.

We would like to thank Stefan Berg from the Bayerische Landesanstalt für Weinbau und Gartenbau, Veitshöchheim, for his expert support and provision of honey bee colonies for the experiment, Susanne Schiele for her excellent assistance during the fieldwork, and Patrick Kohl for recording waggle dance runs. We thank Emily Poppenborg, Sarah Redlich, and the reviewers for valuable comments on the manuscript. The Bundesamt für Kartographie und Geodäsie kindly provided access to land use data on the study site.

Additional Information and Declarations

Competing Interests

Author Contributions

Data Availability

The authors declare there are no competing interests.

Fabian Nürnberger conceived and designed the experiments, performed the experiments, analyzed the data, contributed reagents/materials/analysis tools, wrote the paper, prepared figures and/or tables.

Ingolf Steffan-Dewenter and Stephan Härtel conceived and designed the experiments, contributed reagents/materials/analysis tools, wrote the paper, reviewed drafts of the paper.

The following information was supplied regarding data availability:

The raw data has been supplied as a Supplementary File.

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
