# Peer review of "Combined effects of waggle dance communication and landscape heterogeneity on nectar and pollen uptake in honey bee colonies"

_PeerJ, doi:10.7717/peerj.3441_

## Round 0.1 · original submission · Major Revisions

As you can see, both reviewers found your work valuable and interesting. Both also suggested additional experiments, although they were somewhat vague as to whether these could be also reported in later publications. Of course, it is always more elegant to answer experimental confounds directly, experimentally, in the same publication. However, given the seasonal restrictions on bee research and the constraints we all live in, there likely is the possibility of making the confounds more explicit and discussing their consequences, even if experiments would be more desirable. Hence, I suggest if it is at all possible, to address the issues raised by the reviewers experimentally. If, however, you have good reasons that force you to refrain from that option, do make these reasons explicit in your rebuttal letter.

Reviewer 1 ·

Basic reporting

This ms reports a set of interesting data addressing the question whether landscape structures effect the foraging success of honeybee colonies with or without using social communication via the waggle dance. The literature on the specific topic (ecology of honeybee foraging with or without disturbance of waggle dance performance) is well covered in the Introduction and the Discussion. However, the broader literature on waggle dance communication is more or less ignored. This is unfortunate, because the authors would have found that an important notion had been introduced in this literature distinguishing between the motivational and the instructive components of the waggle dance (e.g. [1], [2]). The concept of “private” and “social” information is not very helpful in this context (see specific points).

Experimental design

The design of the experiments is based on a clear hypothesis and meets the requirements given the problem that the heterogeneity of landscape structure with respect to food distribution (nectar, pollen) is very difficult to quantify. The attempts of the authors are commendable in this respect. For future studies the distance between patches should be quantified and representative samples of nectar production should be provided. Several problems arise from the methods applied and the limited (practically not existing) data about the performance of waggle dances. Furthermore, it would have been necessary to evaluate whether the frequent (every 4 days) move of the colony to a new site and a turn of the hive from a vertical to a horizontal condition compromises colony conditions as measured with more sensitive methods than with relative weight change and pollen transport. Additional colonies should have been left in the respective environments in order to test how prolonged exposure to the particular landscape effects foraging success. It is also unfortunate that colonies lost weight over the time of observation rather than gaining weight indicating that foraging success was measured under very specific ecological conditions, a topic that should be addressed more critically in the discussion.

Validity of the findings

The data add to the existing literature, but since most of the findings do not support the hypothesis the findings are particularly important. The specific comments will show in which direction any follow-up study should be developed.

Additional comments

I suggest that the authors should also receive my comments above in the threes boxes.

Specific points
l. 25: variation is misleading, better differences. The word temporal is also misleading because you are not studying the temporal effect of change in the different distributions of food patches
l. 30 weight gain is misleading because you did not find any gain, better: weight change or weight loss
l 88 The term heterogeneity is used throughout the ms without reference to the geometric arrangements of landmarks guiding bees in their foraging flights. Since the colonies are translocated to different sites foraging bees will apply their memory about the formerly learned landscape to orient in the new landscape. Therefore, the question arises whether and how the landscapes differ with respect to the geometric arrangement of guiding landscape structures (panorama, elongated ground structures, etc). The effect of the waggle dance for recruitment might heavily depend on the amount of similarity or differences in these respects between the selected environments.
l. 96 – 99 What is meant with this sentence? The effect may just be opposite: the knowledge from the former environments may facilitate and enhance the effect of dance communication particularly if the general layout of the environments have overlapping features that are transferred between the environments making it possible for the bee to gain from the motivational component of the waggle dance (see general points and below) without the instructive component (e.g. on the horizontal comb).
L- 118 A distance of 5 km is certainly too short if one wants to exclude the possibility of familiarity with at least parts of the environment. Were tests performed to find out whether bees returned to the old site?
l. 120 you need to say that the features of the landscape were analyzed within an area of 2 km radius: not the landscape had a radius…
l. 126 -147 l. 147 the most important parameter is missing here, the distance (and variance of distances) between the patches. This is the parameter which was referred to (intuitively) by Seeley and Chittka.
l. 170 Were these videos quantified, and how many recordings were performed? I get the impression that these were not systematic recordings in the colonies during their horizontal placement. The problem is that also the cited literature is rather imprecise in this respect.
l. 188 this is very unlikely. Bees learn to relate the landscape structures to their sun compass system during their first exploratory orientation flights. They transfer this knowledge to a new site ([3], , [4, 5]). See also my comment above and below about the effect of the motivational component in dance communication.
l. 200 better: weight loss. It took me quite a time to notice that colonies lost gain. Therefore, it will be helpful to the reader not to induce an expectation that the colonies had gained weight throughout the experiment (although the term weight gain is a technically introduced term).
l. 234 It might still be informative to compare the effects pairwise and I leave it for you to search for a procedure that allows for such a pairwise comparison
l. 139 Neither Table 1 nor Table 2 give any measure of nectar/pollen productivity of the flowers. What is the measure of “scarcity of floral resources”?
l. 280 as mentioned already several times bees are not landscape-naïve when moved to a new environment. They generalize from their former experience with an environment to the new environment. Therefore, assuming that they would behave in a new environment in the same way as during their first exploratory flights is certainly not supported by data and highly unlikely.
l. 282 Transfer to a new environment also does not reflect conditions in the same environment after an interruption due to bad weather conditions. Bees have a long lasting memory for landscape structures and feeding places lasting even for several months ([5]).
l. 284 I already mentioned that the spatiotemporal (why temporal?) parameters used here to define heterogeneity is so remote from the (intuitive) measures of the authors cited here. Therefore, it makes very little sense to make comparisons between the data presented here and that of these authors.
l. 297 The term “private information” (if used at all, see general comment) needs to be defined here. Is private information also the learned odor or nectar taste communicated during the waggle dance? Certainly not. Why should bees dancing on a horizontal comb not communicate this information?
l. 302 this is not correct: you have not prevented bees from using previously acquired private information. This applies to several aspects: generalization between landscapes (see above), learned odors and tastes in the former environments, specializing on nectar or pollen foraging, etc.
l. 314 This sentence is not supported by data. There are not specific data on this point in the book by von Frisch or his other publications.
l. 319 this should have been measured
l. 325 This conclusion needs to include the missing controls and data as mentioned before. The next sentence give parts of these points. It will be better to give the sentence l. 326 – 327 first.
l. 330 see my comments above with respect to the motivational component of dancing and “private information”. This argument should not be introduced at the very end of the paper because it is an essential limitation of the whole study and should not be obscured. It needs to be included in the Abstract and the general conclusion.
l.338 to which references are referred to here?
l. 342 – 343 include two “may”: … and hence may . This may reduce…
l. 347 to which references do you refer here?


References

1. Menzel, R., Kirbach, A., Haass, W.-D., Fischer, B., Fuchs, J., Koblofsky, M., Lehmann, K., Reiter, L., Meyer, H., Nguyen, H., et al. (2011). A common frame of reference for learned and communicated vectors in honeybee navigation. Curr Biol 21, 645-650.
2. Seeley, T.D. (2011). Olfactory Information Transfer During Recruitment in Honey Bees. In Honeybee Neurobiology and Behavior, Volume 1, C.G. Galizia, Eisenhardt, D. Giurfa, M., ed. (Heidelberg, New York: Springer Verlag).
3. von Frisch, K., and Lindauer, M. (1954). Himmel und Erde in Konkurrenz bei der Orientierung der Bienen. Naturwiss 41, 245-253.
4. Dyer, F.C., and Gould, J.L. (1981). Honey bee orientation: A backup system for cloudy days. Science 214, 1041-1042.
5. Lindauer, M. (1970). Lernen und Gedächtnis - Versuche an der Honigbiene. Naturwiss 57, 463-467.

·

Basic reporting

This article explores an interesting and much debated question, namely whether and when the honeybee dance language helps colonies to collect more food. Several studies have addressed this question in the last years and the findings of the studies suggest that communicating vector information improves foraging performance only in certain conditions. In particular, it is currently not known if vector information improves foraging success in temperate European habitats. The authors test the hypothesis that vector information improves colony foraging success in temperate habitats when resources are scarcer and landscapes are more heterogeneous. The writing of the article is clear and the language easy to understand. One important issue is that, as in previous studies, the authors do not test the importance of the waggle dance, but the importance of the spatial/vector information of the waggle dance. As the authors discuss later, the vector information is just one component of the waggle dance communication and possibly not even the most important one (see e.g. Grüter & Farina 2009, Trends Ecol. Evol.). Hence, it is important to make a clear distinction between the waggle dance and the vector information. Otherwise, many statements become misleading, as is the case with the first sentence of the abstract.
Abstract
- L22: Make a clear distinction between waggle dance and the vector component. Furthermore, tropical habitats are very diverse and, as far as I know, the role of vector information has been studied in just one tropical habitat with 2 colonies. Thus, I would be careful not to make general statements about what you call tropical systems.
Introduction
- In my opinion the introduction reduces social foraging in honeybees too much to the vector information of the waggle dance. I would argue that other information components of the waggle dance are similarly important and should be mentioned, e.g. food odours, cuticular hydrocarbons, the interaction with private information. This is shortly mentioned in the discussion. Furthermore, only about 10% of all foragers dance, but close to 100% perform trophallaxis, i.e. they offer food samples (or receive them) to nest-mates. Foragers acquire important information during trophallaxis (food odour and profitability), which affect their foraging decisions (see the work of Walter Farina and collaborators). For example, already young bees learn food odours during trophallaxis and they use this information to decide which dances to follow or on which plants to land (Arenas et al. 2007, 2008; Balbuena et al. 2012). Thus, there is constant exchange of information even in the absence of waggle dancing. I fully understand that the authors want to mainly focus on the vector information, but the richness of information sources that are used in foraging should be mentioned shortly. Otherwise the representation of social information used in foraging in the introduction comes across as a bit simplistic.
- L70: it is not clear whether the dance communication indeed evolved in the tropics. Since the Dyer & Seeley 1989 paper was published, several Apis fossils have been discovered in Europe and it has been suggested the Apis and the dance language might have evolved in Europe in a more temperate climate (Kotthoff et al. 2013, ZooKeys; I’Anson-Price & Grüter 2015, Frontiers in Ecol. Evol.).
- L72: I suggest removing the “simple”.

Figures
- The figures summarize the findings well, even though I suggest removing best fit lines if the relationships are not significant, as in Fig. 3. Als
- The only figure I struggled with was the supplementary figure. I didn’t see sample sizes or details on how data was collected. For example, how many waggle runs per dance and how many dances were analysed? Why do you show vector length, rather than the more commonly used angular deviation of dances to show that the dances were disoriented (e.g. Sherman & Visscher 2002)?

Experimental design

- The methods of the study show some important improvements compared to earlier work, in particular the larger number of colonies and the prevention of confounding memory effects. The questions are well defined and relevant. However, I also think that the study has a couple of serious methodological problems:
- Most importantly, the two experimental groups differ in two aspects, 1) the presence of either disoriented or oriented dances and 2) the orientation of the combs (horizontal vs. vertical). You are interested in the effect of 1), but cannot exclude that any effects or the absence of effects of your treatment is caused by 2). Thus, your findings regarding pollen collection could be explained by the comb orientation, rather than the dance orientation. There are many possible reasons why this could be the case, one is that brood in the unnatural horizontal comb position produced less hunger signal during the period that you studied, thereby reducing the motivation of foragers to collect pollen (this does not apply to nectar foraging because nectar foraging does not depend on the brood signal). In the discussion you try to argue that this is unlikely, but there is no way of knowing whether your results are caused by effects of dance orientation or differences in the motivation to collect pollen caused by comb orientation. Another question is whether bees performed fewer dances on horizontal combs than vertical combs. This could have different effects on pollen and nectar foraging. Did you quantify dance motivation in the two conditions to exclude this possibility? I think that this methodological aspect of the study makes it difficult to interpret your findings without further experiments. The problem that you don't actually know what caused your pollen-effects should be acknowledged in the discussion.
- Another problem I have is that you compared pollen foraging on just one day shortly after moving colonies to a new location, but it is not clear how these findings can be extrapolated to more days since experimental time windows have a big impact on the benefits of dance communication (Schürch & Grüter 2014).
- Finally, the experimental design represents an unnatural situation: colonies are moved every 4 days to completely new locations. This means that foragers first need to learn the new location of the colony, learn landmarks and when they have discovered good food sources by following dances they cannot benefit from this information for a long time because they are moved to a new location. You argue that this is similar to a natural situation with interruptions by bad weather but I have serious doubts that this is the case. Even after a few days of rain the environment does not change nearly as dramatically and as regularly as in your study. Simulation studies show that vector information is more beneficial in stable environments (Schürch & Grüter 2014), which suggests that your colonies were unable to fully benefit from vector information.
- It’s not clear to me how you determined whether the flowers you recorded were indeed good food sources for honeybees. For example, as far as I know white clover is a better food source for honeybees than red clover due differences in corolla length. How did measure whether the flowers in your study were actually profitable food sources for honeybees?
- Please specify whether all of your 24 colonies were tested in either conditions or only some of them. If you tested them in both conditions (L301), how did you determine the order of the conditions?

Validity of the findings

- Your finding that there is no effect on nectar collection is maybe not so surprising given that you moved the colonies every 4 days. This represents a rather short-lived environment. Schürch & Grüter (2014) show that vector information is not very beneficial in rapidly changing environments.
- Could you offer a mechanistic explanation for your results? Why should dancing for pollen sources be more important than for nectar sources? Do you expect bees to follow nectar dances less or for shorter time periods? Are they less able to decode the nectar dances and, therefore, dancing is not as important? Do bees dance less for nectar than for pollen?
- L281-282: I doubt that bad weather causes such drastic environmental changes. For example, bees will continue to visit the same field of oil-seed rape or fruit orchard even after a few days with rain.
- L319-321: I do not doubt that this statement is correct, but we still do not know whether comb orientation affects the production of hunger signal by the brood during the short period you studied or whether horizontal comb orientation affects the motivation to collect pollen in other ways. Without additional observations (e.g. horizontal combs with oriented dances) it is hard to know what caused the reduction in pollen collection.
- L328-329: I think von Frisch 1967 (1968) would be a better citation here.
- L349-350. I think that such a strong statement is not justified given that you changed the environment the colonies experienced so frequently.

---

## Round 0.2 · Minor Revisions

Only one of the reviewers had a few small comments. I do not see any problems addressing them.

Reviewer 1 ·

Basic reporting

the Revision of the MS has increased considerably the clearness of the reportung

Experimental design

is ok within the limits that become much clearer now after revision

Validity of the findings

important findings well presented in the revised ms

Additional comments

I am happy with the way the authors dealt with the comments and the changes they have made accordingly.

·

Basic reporting

- L53: maybe use Apis instead of Apis mellifera, since there are about 10 Apis species that perform waggle dances.
- Figure 5: normally one assumes that there was a significant difference between intact and disrupted colonies if you show two different lines in your graph. If there is no difference, then you should show just one line representing the mean line for both groups.
- Figure 6: the text states that there was no interaction between patch size and treatment, hence you shouldn’t show two lines with different slopes as this indicates an interaction.
- L306: dances do not only activate idle foragers, but also reactivate experienced foragers that are temporarily unemployed. Most dance following seems to be of this kind, given how brief dance following often is (Grüter & Farina 2009, Trends Ecol. Evol.).
- L410: here you could also consider Couvillon et al. 2014, Curr. Biol.

Experimental design

see general comments

Validity of the findings

- L294-296: I do not understand this statement. Are you referring to Figure S2? However, this does not measure brood rearing activity or, more importantly, brood signalling activity during the time period when you collected the pollen data nor does it compare intact with disrupted colonies.
- L357-361: This is a rather general statement and I would suggest adding the caveat that your statement is valid when looking at short-term benefits because your experimental design only measured short-term benefits. In the longer-run, different patterns might emerge.
- L392-393: The problem with the data shown in Fig. S2 is that there is no comparison with the horizontal situation. This makes it impossible to judge whether the treatment affected brood rearing. Thus, more research is needed to elucidate this point. Maybe the authors could highlight this as an area for future research.

Additional comments

The authors have satisfactorily addressed most of my concerns. The main remaining point that might require attention is the effect the horizontal treatment might have on brood rearing and signalling. Even by adding new information (Fig. S2) you cannot rule out that the treatment affected brood behaviour in the short-term, thereby affecting pollen collection. Regarding this issue, I think that the statement on L294-296 is a bit misleading because you didn’t compare brood rearing in intact vs. disrupted colonies. And even if this data were available, I don’t think it would answer the question. Instead, the authors could suggest that further work might help differentiate between the two explanations.
Here are a few minor comments:

---

## Round 0.3 · accepted · Accept

The final reviewer has indicated that they are satisfied with your revisions and that your article is now ready for publication.
Congratulations!